

# Anomalous phase ordering
# of a quenched ferromagnetic superfluid

## Lewis A. Williamson[1,2*] and P. Blair Blakie[1]

**1** Dodd-Walls Centre for Photonic and Quantum Technologies,
Department of Physics, University of Otago, Dunedin 9016, New Zealand
**2** Department of Physics, Lancaster University, Lancaster, LA1 4YB, United Kingdom

⋆ l.a.williamson@lancaster.ac.uk

## Abstract

Coarsening dynamics, the canonical theory of phase ordering following a quench across a symmetry breaking phase transition, is thought to be driven by the annihilation of topological defects. Here we show that this understanding is incomplete. We simulate the dynamics of an isolated spin-1 condensate quenched into the easy-plane ferromagnetic phase and find that the mutual annihilation of spin vortices does not take the system to the equilibrium state. A nonequilibrium background of long wavelength spin waves remain at the Berezinskii-Kosterlitz-Thouless temperature, an order of magnitude hotter than the equilibrium temperature. The coarsening continues through a second much slower scale invariant process with a length scale that grows with time as $t^{1/3}$. This second regime of coarsening is associated with spin wave energy transport from low to high wavevectors, bringing about the the eventual equilibrium state. The transport displays features of a spin wave energy cascade, providing a potential profitable connection with the emerging field of spin wave turbulence. Strongly coupling the system to a reservoir destroys the second regime of coarsening, allowing the system to thermalise following the annihilation of vortices.


# 1  Introduction

Quenching a system across a continuous phase transition from a high to low symmetry phase causes the system to spontaneously break symmetry. Immediately after the quench causally disconnected regions of the system will break symmetry independently, resulting in the formation of domains with independent order parameter orientation. The subsequent growth of these domains toward the global equilibrium state is known as coarsening dynamics. Although the microscopic details of coarsening are usually extremely complicated, at a macroscopic level a much simpler scaling regime can emerge for large average domain size $L$. Spatial correlations of the order parameter at different times $t$ then collapse onto a single curve when rescaled by $L$, and the domains grow as $L \sim t^{1/\eta}$ with the scaling exponent $\eta$ determined by the dynamic universality class [1]. Such universal dynamics has been explored in a vast variety of systems, ranging from the early universe [2] to superfluid formation [3] to opinion spreading in sociology [4]. When the quench produces topological defects, the decay of these defects has long been thought to provide a unifying framework for understanding the coarsening [1].

Recently, there has been much interest in coarsening dynamics in ultracold atom systems, which are well isolated from their environment and present a pristine system for studying nonequilibrium phase transitions [5–13]. Of particular interest are multicomponent condensates, which support a rich variety of order parameter manifolds and associated topological defects [14,15]. Theoretical studies of coarsening in a variety of cold atom systems [3,16–28] have culminated in the recent experimental observation of universal dynamics in a quenched quasi-1D scalar Bose gas [6] and in a quenched quasi-1D spin-1 condensate [7]. Simulations of a homogeneous quasi-2D spin-1 condensate quenched from the polar phase to the easy-plane ferromagnetic phase, see Fig. 1(a), identified coarsening dynamics driven by the mutual annihilation of transverse spin vortices with domain size growing as $L \sim t/\log t$ [19,20]. A log correction to scaling is familiar from two dimensional systems supporting vortices [1].

In this work we study the easy-plane ferromagnetic ordering of a homogeneous quasi-2D spin-1 condensate after all vortices have annihilated. Remarkably, we find that the annihilation of vortices does not take the system to the equilibrium state. Instead, a nonequilibrium background of spin waves remain at the Berezinskii-Kosterlitz-Thouless (BKT) temperature, an order of magnitude hotter than the eventual equilibrium temperature. The coarsening then continues via spin wave energy transport from low to high wavevectors that displays features of a novel turbulent cascade, relevant to the emerging area of spin turbulence (e.g. see [13,29–32]). We argue that the nonlinear transverse spin wave dynamics arises from a dynamic coupling to interacting axial spin degrees of freedom. Order parameter correlations show dynamic scale invariance during the spin wave coarsening, with a length scale that grows as $t^{1/3}$. This scaling is distinct from that during the vortex driven coarsening, showing that there are two renormalisation group fixed points affecting the phase ordering of this system.

Strongly coupling the system to a reservoir of energy and particles destroys the second scaling regime, allowing the system to thermalise following the annihilation of vortices. Our results give new insights into the phase ordering dynamics of isolated systems and provide a potential profitable connection between phase ordering and wave turbulence.

## 2 Background

A spin-1 condensate can be described by three interacting classical fields $\psi_m$ for condensates in the three spin components with spin projections $m = -1, 0, 1$. The quasi-2D Hamiltonian density within a uniform trap [11] is [33–36],

$$\mathcal{H} = -\sum_{m=-1}^{1} \psi_m^* \frac{\hbar^2 \nabla^2}{2M} \psi_m + \frac{g_n}{2} n^2 + \mathcal{H}_s,  \tag{1}$$

where $M$ is the atom mass, $n = \sum_m |\psi_m|^2$ is the areal density, $g_n$ is the quasi-2D density interaction strength and $\mathcal{H}_s$ encompasses the spin dependent terms,

$$\mathcal{H}_s = \frac{g_s}{2} n^2 |\mathbf{F}|^2 + \sum_{m=-1}^{1} q m^2 |\psi_m|^2.  \tag{2}$$

The first term in $\mathcal{H}_s$ is the spin interaction energy, with spin density $\mathbf{F} = \sum_{mm'} \psi_m^* \mathbf{f}_{mm'} \psi_{m'} / n^2$ for spin-1 matrices $(f_x, f_y, f_z) \equiv \mathbf{f}$, and quasi-2D spin interaction strength $g_s$. The sign of $g_s$ determines whether the interactions are ferromagnetic ($g_s < 0$), which occurs in $^{87}$Rb [37], or antiferromagnetic ($g_s > 0$), which occurs in $^{23}$Na [35]. Here we consider the ferromagnetic case. The second term in $\mathcal{H}_s$ is a quadratic Zeeman splitting of the spin components, which can be induced using either DC magnetic fields or AC microwave stark shifts [15,38]. A linear Zeeman term $pnF_z$ can also be included, but conservation of $nF_z$ means this term does not affect the system dynamics and can be removed via the unitary transformation $e^{-ipmt/\hbar} \psi_m \to \psi_m$. The quasi-2D regime is obtained from a 3D system by tightly confining the system in one direction and integrating over the resulting spatial profile along that direction [5,36].

The relative strength of the two terms in $\mathcal{H}_s$ produces a rich phase diagram, from which a variety of quenches can be explored. The zero temperature mean field phase diagram for ferromagnetic interactions and with $q > 0$ is shown in Figure 1(a). A quantum critical point at $q = q_0 \equiv 2|g_s|n_0$ ($n_0$ is the mean condensate density) separates the unmagnetised polar phase (all atoms in the $m = 0$ condensate) from the easy-plane ferromagnetic phase with spin order parameter $\mathbf{F}_\perp \equiv (F_x, F_y)$ (for quantization along $F_z$). The order parameter manifold of $\mathbf{F}_\perp$ is SO(2) with transverse spin vortices as topological defects. These vortices consist of a positive or negative phase winding of the transverse spin angle $\theta$ ($\tan\theta = F_y/F_x$), and can only decay via the mutual annihilation of two vortices of opposite sign. Vortices with negative phase winding are also termed antivortices. The energy scale $q_0$ defines a time scale $t_s \equiv \hbar/q_0$ and the spin healing length $\xi_s \equiv \hbar/\sqrt{Mq_0}$.

## 3 Results

### 3.1 Quench dynamics: anomalous phase ordering

We simulate the condensate dynamics following an instantaneous quench of the quadratic Zeeman energy from deep in the polar phase to $q = 0.3q_0$ in the easy-plane ferromagnetic phase, see Fig. 1(a),(b). Symmetry breaking and the production of transverse spin vortices following

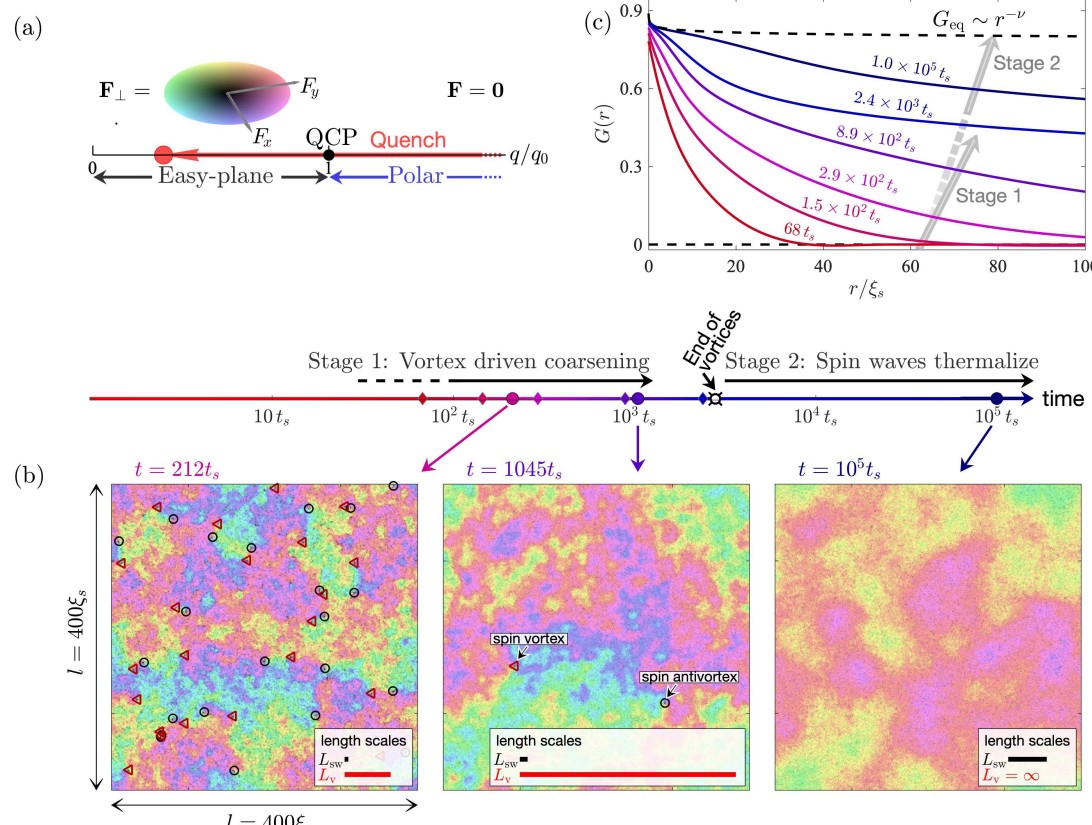

Figure 1: (a) A spin-1 ferromagnetic condensate is unmagnetised ("polar") for $q > q_0$ and magnetises in the transverse plane ("easy-plane") for $0 < q < q_0$. The point $q = q_0$ is a quantum critical point (QCP). We explore the ordering of transverse spin following a quench from $q \gg q_0$ to $q = 0.3q_0$. (b) Coarsening of transverse spin domains [colormap shown in (a)]. This is associated with collisions between transverse spin vortices (red triangles) and antivortices (black circles), which can then mutually annihilate, resulting in a growing intervortex spacing $L_v$ (red bar). A second length scale $L_{sw}$ giving the thermal wavelength of spin waves grows much more slowly (black bar), such that the transverse spin remains out of equilibrium long after all transverse spin vortices have annihilated. The central time axis quantifies the first stage of vortex driven coarsening, the time of last vortex annihilation, and the subsequent stage of spin wave thermalisation. (c) Spatial correlations of transverse spin at different times, showing that long after all vortices have annihilated the correlations still decay more rapidly than the equilibrium prediction (upper black dashed line).

such a quench have been observed in experiments with $^{87}$Rb [5]. Conservative dynamics of our system is simulated by numerically integrating the three coupled Gross-Pitaevskii equations (GPEs) obtained from Eq. (1) [14],

$$i\hbar \frac{\partial \psi_m}{\partial t} = \left( \frac{\hat{p}^2}{2M} + qm^2 + g_n n \right) \psi_m + g_s n \sum_{m'} \mathbf{F} \cdot \mathbf{f}_{mm'} \psi_{m'}. \tag{3}$$

Further numerical details are described in Appendix A.1. A homogeneous system can be realised in experiments using a flat bottomed trap [11, 39].

We quantify order in the system by spatial correlations of $\mathbf{F}_\perp$,

$$G(r, t) \equiv \langle \mathbf{F}_\perp(\mathbf{r}, t) \cdot \mathbf{F}_\perp(\mathbf{0}, t) \rangle, \tag{4}$$

where angular brackets denote an ensemble average (see Appendix A.2). Figure 1(c) shows the evolving correlation function Eq. (4). For times $10^2 t_s \lesssim t \lesssim 10^3 t_s$, the growth of order is scale invariant and driven by the mutual annihilation of transverse spin vortices of opposite sign, see Fig. 1(b), with correlations decaying to zero at a length scale on the order of the intervortex spacing. This vortex driven coarsening has been described in previous work [19, 20]. We find that all vortices have annihilated by a time $t \approx 2.8 \times 10^3 t_s$, after which correlations extend to the boundary. The correlations can then be compared to the equilibrium (thermalised) prediction [36, 40],

$$G_{\text{eq}}(r) \sim r^{-\nu}, \qquad \nu = \frac{T_{\text{eq}}}{4 T_{\text{BKT}}}. \tag{5}$$

Here $T_{\text{BKT}} = \pi K / 2 k_B$ is the BKT temperature associated with the unbinding of transverse spin vortices [41], with $K = \hbar^2 n_0 (1 - q/q_0)/2M$ the spin wave stiffness and $k_B$ Boltzmann's constant. The equilibrium temperature $T_{\text{eq}}$ of our microcanonical system is calculated by equipartitioning the energy liberated by the quench amongst all collective modes of the system [20], which gives $\nu \approx 0.011$. This equilibrium prediction is shown in Fig. 1(c). Surprisingly, even after very long simulation times $t = 10^5 t_s$, correlations of transverse spin only agree with the equilibrium prediction for length scales $r \lesssim 5\xi_s$. For larger length scales the correlations decay more rapidly. This absence of equilibrium following the annihilation of topological defects is not predicted by the current theory of coarsening dynamics [1].

## 3.2 Spin wave energy transport driving phase ordering

To identify the origin of the unexpectedly slow ordering displayed in Fig. 1(c) we look at the distribution of energy in the gradient of the transverse spin angle $\nabla \theta$ (this vector field is proportional to currents of $F_z$ magnetization [42]). We firstly perform a Helmholtz decomposition $\nabla \theta = \mathbf{v}_i + \mathbf{v}_c$ with $\nabla \cdot \mathbf{v}_i = 0$ and $\nabla \times \mathbf{v}_c = 0$. The first contribution $\mathbf{v}_i$, known as the incompressible field, arises from vortex excitations while the second contribution $\mathbf{v}_c$, known as the compressible field, arises from transverse spin wave excitations. The spectral energies of the incompressible and compressible fields are given by,

$$\epsilon_\mu(k, t) = \frac{K}{2} \left\langle \left| \tilde{\mathbf{v}}_\mu(\mathbf{k}, t) \right|^2 \right\rangle, \qquad \mu = \text{i, c}, \tag{6}$$

where $\tilde{\mathbf{v}}_\mu(\mathbf{k}) = l^{-1} \int d^2 \mathbf{r} \, \mathbf{v}_\mu(\mathbf{r}) e^{-i \mathbf{k} \cdot \mathbf{r}}$ is the Fourier transform of $\mathbf{v}_\mu(\mathbf{r})$ and angular brackets denote an ensemble average (see Appendix A.2).

The evolving spectral energies $\epsilon_\mu(k, t)$ are shown in Fig. 2(a),(b). The incompressible spectral energy, Fig. 2(a), shows a $k^{-2}$ decay when vortices are present, in agreement with the infrared ($\xi_s k < 1$) scaling of a distribution of quantum vortices [43, 44]. Once all vortices have annihilated the spectral energy drops abruptly. In comparison, the compressible spectral energy, Fig. 2(b), shows nonequilibrium features across the duration of the simulation. The initial condition of our simulation results in a flat high energy distribution $\epsilon_c(k, 0) \approx 200 k_B T_{\text{eq}}$. For times $t \gtrsim 10^3 t_s$, the compressible spectral energy shows three approximate regimes,

$$\epsilon_c(k, t) = \begin{cases} \epsilon_{\text{lw}}(k, t), & k < k_{\text{lw}}(t), \\ \left( \epsilon_{\text{lw}}(k_{\text{lw}}, t) / k_{\text{lw}}^{-\alpha} \right) k^{-\alpha}, & k_{\text{lw}}(t) \le k < k_{\text{eq}}(t), \\ k_B T_{\text{eq}} / 2, & k_{\text{eq}}(t) \le k. \end{cases} \tag{7}$$

We have introduced the evolving wavevectors $k_{\text{lw}}(t)$ and $k_{\text{eq}}(t)$ to signify the boundaries between the three regimes of $\epsilon_c(k, t)$. The spectral energy $\epsilon_{\text{lw}}(k, t)$ is the long wavelength portion of $\epsilon_c(k, t)$, with energy per mode $\epsilon_{\text{lw}}(k, t) \approx 10 k_B T_{\text{eq}} \approx k_B T_{\text{BKT}}/2$ in the wavevector window

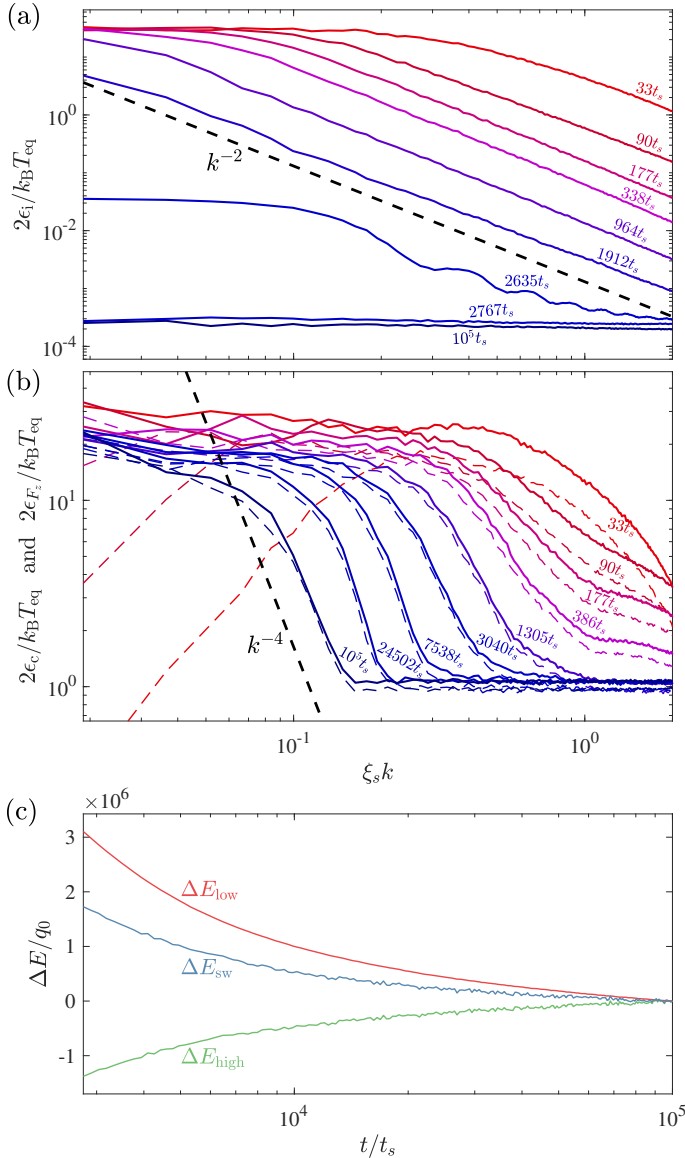

Figure 2: (a) The evolving incompressible field spectral energy $\epsilon_i(k, t)$ displays a predicted $k^{-2}$ scaling and rapidly drops after all vortices have annihilated. (b) The evolving compressible field spectral energy $\epsilon_c(k, t)$ (solid lines) shows three regions: a persistent high temperature long wavelength region with a temperature approximately equal to $T_{\mathrm{BKT}}$; a steep region with an approximate $k^{-4}$ scaling; and a short wavelength thermal region. The spectral energy of $F_z$ excitations $\epsilon_{F_z}(k, t)$ (dashed lines) closely follows $\epsilon_c(k, t)$ for times $t \gtrsim 400t_s$. The interacting $F_z$ fluctuations can mediate the thermalisation of the transverse spin waves. (c) The combined spectral energy of transverse and axial spin waves, $E_{\mathrm{sw}}(t)$, is decomposed into a low wavevector portion $E_{\mathrm{low}}(t)$, which decreases in time, and a high wavevector portion $E_{\mathrm{high}}(t)$, which increases in time, consistent with a cascade of energy from low to high wavevectors. The total spin wave energy $E_{\mathrm{sw}}(t)$ also decreases in time indicating energy flow away from spin waves.

considered. This nonequilibrium temperature, being approximately at the BKT temperature, corresponds to the typical energy of a single transverse spin vortex [41, 45], and may be a remnant of interactions between spin waves and vortices during the vortex driven coarsen-

ing. For $k > k_{\text{lw}}$ $\epsilon_c(k,t)$ decays steeply as $k^{-\alpha}$ with an exponent $\alpha \approx 4$ until the equilibrium distribution $\epsilon_c(k,t) = k_B T_{\text{eq}}/2$ is reached at a wavevector $k_{\text{eq}}(t)$. The structure of $\epsilon_c(k,t)$ is suggestive of a turbulent cascade, with a high temperature long wavelength energy source cascading to a short wavelength thermal field. We provide further evidence of this shortly. With no vortices present, the persistent nonequilibrium features of $\epsilon_c(k,t)$ must be responsible for the anomalously slow ordering that we observe in Fig. 1(c).

The observed dynamics of $epsilon_c(k,t)$ necessarily involves nonlinear interactions. The field conjugate to the transverse spin phase $\theta$, i.e. the generator of rotations of $\theta$, is $nF_z$, leading to dynamic coupling between transverse spin waves and the axial spin waves of $F_z$ [20, 36]. The interacting axial spin waves can therefore mediate transverse spin wave interactions. (Transverse spin waves can also interact via nematic $N_{zz} = |psi_{-1}|^2 + |\psi_1|^2$ fluctuations [36], however we find $N_{zz}$ fluctuations are thermalised after the vortices have annihilated and are therefore relatively small.) Expanding the system Hamiltonian to quadratic order in $F_z$ and $N_{zz}$ fluctuations [36] gives the spectral energy of axial spin fluctuations,

$$\epsilon_{F_z}(k,t) = n_0 \frac{\hbar^2 k^2/2M + q}{2(1-q/q_0)} \left\langle \left| \tilde{F}_z(\mathbf{k},t) \right|^2 \right\rangle, \tag{8}$$

where $\tilde{F}_z(\mathbf{k}) \equiv l^{-1} \int d^2\mathbf{r}\, F_z(\mathbf{r}) e^{i\mathbf{k}\cdot\mathbf{r}}$ and angular brackets denote an ensemble average (see Appendix A.2). For times $t \gtrsim 400 t_s$ the spectral energy $\epsilon_{F_z}(k,t)$ closely follows $\epsilon_c(k,t)$, see Fig. 2(b), indicating that the dynamics of the two spectra are coupled and in equilibrium with each other. The nonlinear interactions of axial spin waves can allow the redistribution of energy in $\epsilon_{F_z}(k,t)$ and then dynamic coupling to transverse spin waves can actuate the same effect in $\epsilon_c(k,t)$.

To provide further evidence for the presence of an energy cascade in Fig. 2(b) we decompose the total spin wave energy

$$E_{\text{sw}}(t) = \sum_k 2\pi k (\epsilon_c(k,t) + \epsilon_{F_z}(k,t)) \tag{9}$$

into a low wavevector portion

$$E_{\text{low}}(t) = \sum_{k < k_{\text{mid}}} 2\pi k (\epsilon_c(k,t) + \epsilon_{F_z}(k,t)) \tag{10}$$

and a high wavevector portion

$$E_{\text{high}}(t) = \sum_{k \geq k_{\text{mid}}} 2\pi k (\epsilon_c(k,t) + \epsilon_{F_z}(k,t)), \tag{11}$$

where we choose $k_{\text{mid}} = 0.5 \xi_s^{-1}$. In Fig. 2(c) we plot the energy changes $\Delta E(t) \equiv E(t) - E(10^5 t_s)$ of these three quantities for times after all vortices have annihilated. The energy $E_{\text{low}}$ decreases in time while $E_{\text{high}}$ increases, consistent with an energy cascade from $k < k_{\text{mid}}$ to $k \geq k_{\text{mid}}$. There is also a net decrease in the total spin wave energy $E_{\text{sw}}$, showing that energy is also lost from the spin wave excitations, either to other quadratic excitations [36, 46–48] or to excitations beyond quadratic order. In principle, one could solve for the dynamics of these additional excitations to obtain effective spin wave dynamics that would transport energy from low to high wavevectors. Figure 2(b) shows that the spin wave energy transport is associated with an approximate $k^{-4}$ scaling of $\epsilon_c(k,t)$ and $\epsilon_{F_z}(k,t)$. (Note the spectral energies in most studies of 2D turbulence include a $k$ phase space factor so that the $k^{-4}$ scaling observed here would normally be described as $k^{-3}$ scaling.) There are currently no predictions for such a cascade within weak wave turbulence theory [49, 50]. To confirm that the energy transport shown in Figs. 2(b),(c) is a turbulent cascade would require showing that the energy transport is local in wavevector space.

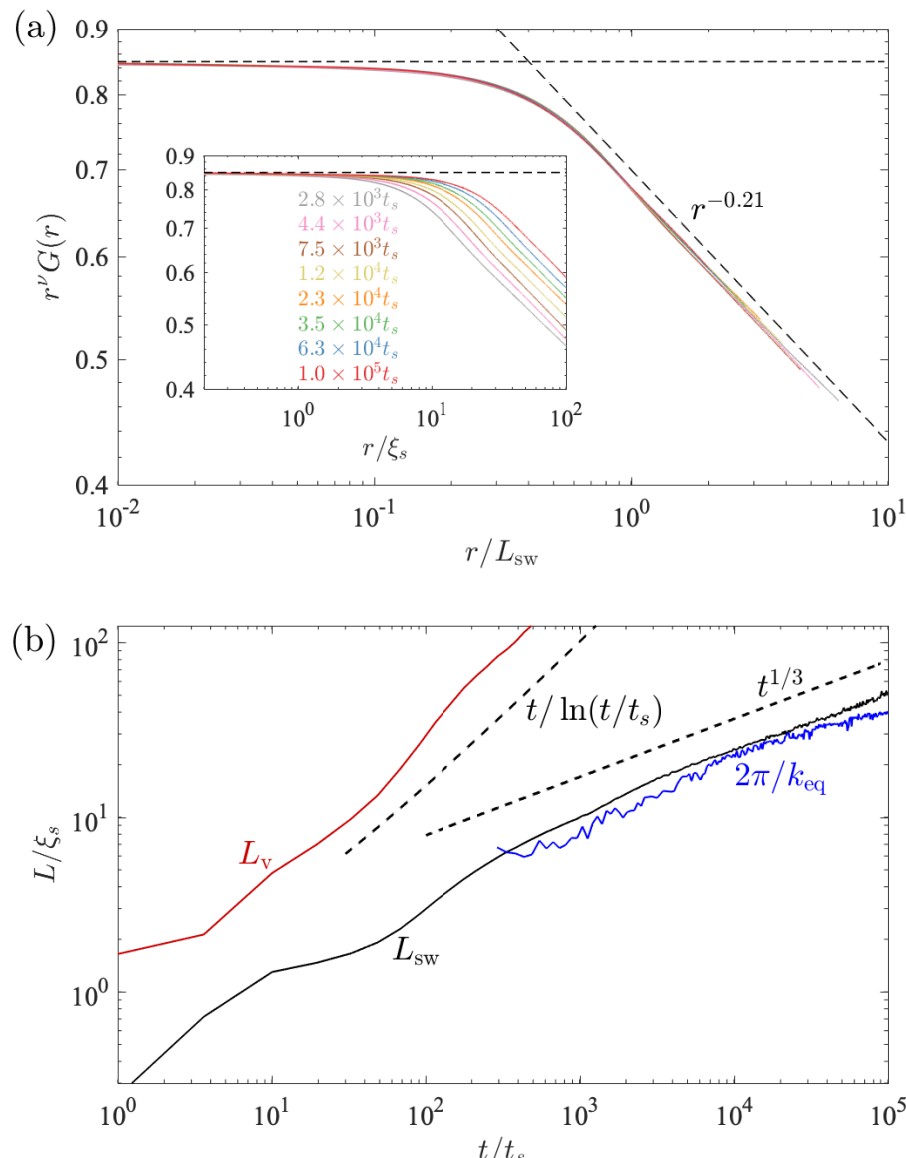

Figure 3: (a) The evolving spatial correlations of transverse spin after all vortices have annihilated (inset) collapse onto a single curve (main figure) according to Eq. (12) when rescaled by the growing length scale $L_{\mathrm{sw}}(t)$. The nonequilibrium (decaying) portion of $r^\nu G(r)$ shows a $r^{-0.21}$ algebraic decay that indicates a nonequilibrium temperature of $T \approx 0.9T_{\mathrm{BKT}}$. The flat dashed line indicates equilibrium correlations. (b) The length scale $L_{\mathrm{sw}}(t)$ grows as $t^{1/3}$ for $t > 10^3 t_s$, much slower than the $t/\ln(t/t_s)$ growth of average intervortex spacing $L_{\mathrm{v}}(t)$. The largest thermarlised wavelength extracted from $\epsilon_{\mathrm{c}}(k, t)$ is $2\pi k_{\mathrm{eq}}(t)^{-1}$, which follows the growth of $L_{\mathrm{sw}}(t)$.

## 3.3 A second regime of scale invariance

The robust shape of $\epsilon_{\mathrm{c}}(k, t)$ for times $t \gtrsim 10^3 t_s$ (see Fig. 2(b)) suggests a regime of scale invariance driven by spin waves, beyond the scale invariant coarsening dynamics driven by vortex annihilation. To explore this we consider the late time dynamics of correlations of

transverse spin, Eq. (4), which in a scale invariant regime will evolve as [51, 52],

$$G(r,t) = r^{-\nu} f\left(\frac{r}{L(t)}\right) \tag{12}$$

for some universal function $f$ and growing length scale $L(t)$. The $r^{-\nu}$ correction factor ensures $G(r,\infty) \sim r^{-\nu}$, consistent with equilibrium. Since $\nu \approx 0.011 \ll 1$, the correction is only significant when $G(r)$ is close to ordered.

The evolving correlation function for times after all vortices have annihilated is shown in the inset to Fig. 3(a). The correlations exhibit a short wavelength ordered portion that grows slowly in time and a nonequilibrium long wavelength portion. The correlation functions collapse onto a single curve after rescaling according to Eq. (12), see Fig. 3(a). We define the rescaling factor $L_{sw}(t)$ by $G(L_{sw}, t) = 0.8 G(0, t)$, which follows the boundary between the ordered portion of the correlation function and the nonequilibrium portion. This length scale is governed by spin waves and grows as a power law $L_{sw} \sim t^{1/3}$ for times $t \gtrsim 10^3 t_s$, i.e. times after all vortices have annihilated, see Fig. 3(b). The length scale $2\pi k_{eq}(t)^{-1}$, where $k_{eq}(t)$ is introduced in Eq. (7) and defined more precisely in Appendix A.3, follows the growth of $L_{sw}(t)$. For comparison, the scale invariance during vortex driven coarsening is associated with the more rapidly growing average intervortex spacing $L_v(t)$ (defined in Appendix A.4) [19]. The nonequilibrium portions of the correlation functions in Fig. 3(a) clearly exhibit an additional algebraic decay $G(r) \sim r^{-0.21-\nu} \approx r^{-0.22}$. The value of the decay exponent corresponds to a temperature of $T \approx 0.9 T_{BKT}$, see Eq. (5), and is consistent with the nonequilbrium temperature of $\epsilon_{lw}(k,t)$ from Eq. (7).

## 3.4 Comparison with open system dynamics

Our analysis so far has considered isolated, energy conserving dynamics. It is of interest to compare our results with open system quench dynamics, where the condensate is coupled to a reservoir of energy and particles. Using a stochastic Gross-Pitaevskii theory (see Appendix A.5), we model a spin-1 condensate strongly coupled to a reservoir with fixed temperature and chemical potential, which we choose such that the equilibrated energy and particle number matches those of the conservative dynamics. We then simulate the same quench as for the isolated system dynamics. Figure 4(a) shows the evolution of transverse spin correlations, Eq. (4), for the open system dynamics. The vortex driven coarsening is comparable to the isolated system case, with correlations showing scale invariant growth. For times after $t \approx 2 \times 10^3 t_s$ correlations in the open system dynamics show excellent agreement with the equilibrium prediction Eq. (5). For comparison, all vortices have annihilated by a time $t \approx 1.8 \times 10^3 t_s$. The results in Fig. 4(a) are in stark contrast to the results in Fig. 1(c) for the isolated system. Indeed, differences in the two cases are apparent from the evolving spin domains, Fig. 1(b) and Fig. 4(b), with the open system being more ordered in the spaces between vortices. For the large reservoir coupling strength we have used here, spin waves in the open system are able to rapidly thermalise directly with the reservoir rather than via interactions with other spin waves. However, we emphasise that microscopically derived reservoir coupling strengths are much smaller than the value we use here [53], and therefore the isolated system dynamics are a realistic approximation to experiments.

The growing length scales $L_v$ and $L_{sw}$ for the open system dynamics, defined as for the conservative dynamics, are shown in Fig. 4(c). The growth of $L_v$ in the open system is very similar to the isolated system growth (denoted by $L_{v,ISO}$ in this figure). In the open system, however, there is no second growing length scale, and $L_{sw}$ follows the growth of $L_v$.

The decay of transverse spin correlations for open system dynamics following quenches to different values of $q$ show good agreement with Eq. (5) once all vortices have annihilated, see

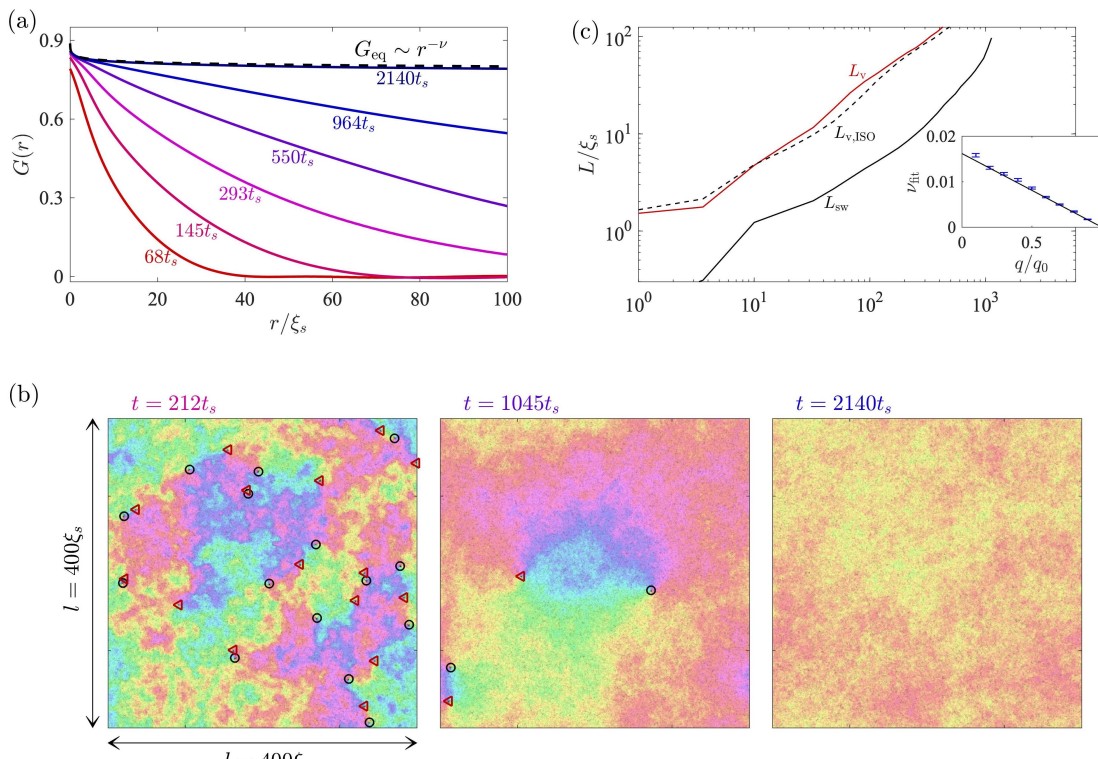

Figure 4: Open system results. (a) Spatial correlations of transverse spin at different times during the coarsening. The correlations agree very well with the equilibrium prediction Eq. (5) (dashed line) after all vortices have annihilated. (b) Coarsening of transverse spin domains [colormap shown in Fig. 1(a)]. Spin vortices and antivortices are marked by red triangles and black circles respectively. Comparing with Fig. 1(b), it is clear that the transverse spin is more ordered in the space between vortices in the open system dynamics. (c) The growth of intervortex spacing $L_v$ follows the isolated system growth $L_{v,ISO}$. The length scale $L_{sw}$ follows the growth of intervortex spacing $L_v$, indicating the absence of a second coarsening process. Inset: Algebraic decay exponent $\nu_{fit}$ for different $q$ quenches (dots) obtained from single trajectory simulations by fitting to the transverse spin correlation function for $2\xi_s \leq r \leq 100\xi_s$ and averaging the result across times $5 \times 10^3 t_s \leq t \leq 10^4 t_s$. The error bars give the standard error of this mean. For $q > 0.1q_0$ the fitted exponents agree well with the equilibrium prediction from Eq. (5) (solid line).

Fig. 4(c) inset. (The temperature and chemical potential for these quenches have been adjusted as a function of $q$; see Appendix A.5.) The small deviation at the smallest $q$ value might be caused by axial spin fluctuations, which become stronger as $q \to 0$ due to a diminishing energy gap. Indeed, we expect that the physics will be modified in the limit $q \to 0$, since the ground state manifold changes from SO(2) × U(1) to SO(3), resulting in changes in collective mode excitations [47, 48] and vortex topology [21].

## 4  Conclusion

We have shown that vortex driven coarsening of an isolated easy-plane ferromagnetic spin-1 condensate does not take the system to equilibrium. Instead, a second regime of scale invariant

coarsening associated with transport of spin wave energy scales more slowly as $t^{1/3}$. Strongly coupling the system to a reservoir of energy and particles destroys this second coarsening process and equilibrium is reached after the vortex driven coarsening.

The presence of two dynamic scaling regimes in the isolated dynamics shows that there are two renormalisation group fixed points affecting the phase ordering. The first, associated with vortices, has been ascribed to the model E dynamic universality class [19]. The second slower scaling, $L_{sw} \sim t^{1/3}$, matches that of the scalar model B dynamic universality class [1, 54], however the order parameter does not: the scalar model B universality class describes a one component conserved order parameter, whereas the order parameter in our system has two components and is not conserved. There is, however, a second important field in our system that does satisfy the properties of the scalar model B universality class: the conserved $nF_z$ field. It could be possible that the dynamics of the $nF_z$ field belongs to the scalar model B dynamic universality class, even though this is not the order parameter of the system, and that the dynamic coupling between $nF_z$ and $\theta$ leads to model B scaling emerging in the correlations of transverse spin. However, model B is also a dissipative model whereas we have shown that strong dissipation destroys the second scaling regime. Hence the second scaling regime might alternatively belong to a currently unidentified dynamic universality class unique to isolated systems. We have shown that this second scaling regime displays features of a spin wave energy cascade, thus identifying a potential connection between the fields of wave turbulence and phase ordering dynamics.

The nonequilibrium background of spin waves that remain after the vortices have annihilated is at a temperature very close to $T_{BKT}$. These spin waves may have thermalised with the vortex field during vortex driven coarsening, either via scattering off of vortices or via spin wave production after vortex annihilation (see [55]). The absence of interactions once all vortices have annihilated would then leave behind high temperature spin waves, reminiscent of photons decoupling from matter in the early universe to produce the cosmic microwave background. This intriguing process may be ubiquitous in phase ordering systems involving topological defects interacting with collective mode excitations.

# Acknowledgements

We thank Dan Stamper-Kurn, Kazuya Fujimoto, Matthew Reeves and Ashton Bradley for valuable discussions.

**Funding information**   We acknowledge support from the Marsden Fund of the Royal Society of New Zealand. LW acknowledges support from the University of Otago Postgraduate Publishing Bursary.

# A   Numerical details

## A.1   GPE simulations

The GPE simulations are conducted on a 2D square grid with dimensions $l \times l = 400\xi_s \times 400\xi_s$ covered by an $N \times N = 512 \times 512$ grid of equally spaced points. In experiments in $^{87}$Rb, $g_n/|g_s| \sim 100$ [37]. We use a more modest ratio $g_n/|g_s| = 10$, which is sufficient to suppress density fluctuations at the energy scale we are interested in. The mean condensate density is taken to be $n_0 = 10^4 \xi_s^{-2}$. We evolve our system using a recently developed fourth order symplectic integrator [56] to ensure that energy, atom number and $nF_z$ magnetization are

conserved effectively. We find that total energy and atom number are conserved to within a factor of $10^{-9}$ across the full simulation time. The total axial magnetization $\int d^2\mathbf{r}\, n(\mathbf{r})F_z(\mathbf{r})$ remains below $10^{-6}n_0 l^2$. We use a time step of $0.02t_s$ for each integration step. The kinetic energy time step is evaluated spectrally using fast Fourier transforms, and we employ periodic boundary conditions. Our initial state is the polar state $(\psi_1, \psi_0, \psi_{-1}) = \sqrt{n_0}(0, 1, 0) + \delta$, where $\delta$ is noise added to Bogoliubov modes on top of the ground state at $q = \infty$, as in [20], which seeds the symmetry breaking evolution. Noise added this way corresponds to adding on average half a particle per mode according to the truncated Wigner prescription [57]. We then evolve our system using Eq. (3) at a quadratic Zeeman energy $q = 0.3q_0$, so that the quench is effectively instantaneous at $t = 0$.

## A.2 Ensemble averaging

Correlations Eq. (4) and spectral energies Eq. (6) and Eq. (8) are computed using an ensemble average of the form,

$$\bar{g}(\mathbf{u}) = \langle g(\mathbf{u}) \rangle, \tag{13}$$

where $\mathbf{u} = \mathbf{r}, \mathbf{k}$ and $g(\mathbf{u})$ denotes the result of a single simulation trajectory. In the GPE simulations, the ensemble average is over 30 simulation trajectories conducted with independent initial noise. In the open system simulations, the ensemble average is over 10 simulation trajectories. We also average $\bar{g}(\mathbf{u})$ over azimuthal angles of the coordinate $\mathbf{u}$, such that $\bar{g}(\mathbf{u}) \to \bar{g}(u)$ for $u \equiv |\mathbf{u}|$. Correlation functions are additionally averaged over space, i.e. we replace $\mathbf{F}_\perp(\mathbf{0}) \cdot \mathbf{F}_\perp(\mathbf{r})$ by $\mathbf{F}_\perp(\mathbf{r}') \cdot \mathbf{F}_\perp(\mathbf{r}' + \mathbf{r})$ in Eq. (4) and average over the spatial coordinate $\mathbf{r}'$.

## A.3 Definition of $k_{eq}$

The wavevector $k_{eq}$ introduced in Eq. (7) is obtained as follows. We firstly skew the spectrum $\epsilon_c(k)$ by multiplying by $k$. We then define $k_{eq}$ as the position of the local minimum that appears in $k\epsilon_c(k)$ at the start of the equilibrium portion of the spectral energy. To improve resolution, we firstly interpolate the numerical values for $k\epsilon_c$ around its minimum and then find the minimum point of the more highly resolved interpolated data.

## A.4 Vortex detection and averaging

We detect vortices by evaluating the phase winding of the transverse spin angle around plaquettes of our simulation grid. The average intervortex spacings in Fig. 3(b) and Fig. 4(c) are defined as $L_v(t) \equiv \left\langle \sqrt{l^2/N_v(t)} \right\rangle$ where $N_v(t)$ is the vortex number for a single simulation trajectory at time $t$ and angular brackets denote an ensemble average over the 30 simulation trajectories for the GPE results and the 10 simulation trajectories for the open system results. The results for $L_v$ in Fig. 1(b) are for the single trajectory displayed.

## A.5 Open system simulations

To model open system evolution we couple our condensate to a reservoir of energy and particles with fixed temperature $T$ and chemical potential $\mu$. The dynamics is simulated using the simple growth stochastic Gross-Pitaevskii equations (SGPEs) [53, 57–59],

$$i\hbar d\psi_m = (1 - i\gamma)(\mathcal{L}_m[\psi_m] - \mu\psi_m)dt + dW(\mathbf{r}, t). \tag{14}$$

Here

$$\mathcal{L}_m[\psi_m] = \left(-\frac{\hbar^2\nabla^2}{2M} + qm^2 + g_n n\right)\psi_m + g_s n \sum_{m'} \mathbf{F} \cdot \mathbf{f}_{mm'}\psi_{m'} \tag{15}$$

is the conservative evolution operator from Eq. (3), $\mu$ is the chemical potential and $\gamma$ is a dimensionless damping. The precise value chosen for $\gamma$ will not affect equilibrium properties, but will affect the rate that equilibrium is approached. The term $dW(\mathbf{r}, t)$ is Gaussian distributed complex noise with delta correlations,

$$\left\langle dW^*(\mathbf{r}, t) dW(\mathbf{r}', t) \right\rangle = \frac{2\gamma k_{\mathrm{B}} T}{\hbar} \delta(\mathbf{r} - \mathbf{r}') dt. \tag{16}$$

The SGPEs Eq. (14) take the form of Langevin equations.

The temperature (as a function of $q$) is chosen to be that obtained by equiparitioning the energy liberated by the quench amongst the $3N^2$ numerical modes, as was done in the calculation of $T_{\mathrm{eq}}$ for the microcanonical system. The energy liberated is the energy of the polar state evaluated at the final quadratic Zeemen energy $q$ [20]. The temperature is then,

$$k_{\mathrm{B}} T = \frac{q_0}{12N^2} \left(1 - \frac{q}{q_0}\right)^2 n_0 l^2. \tag{17}$$

The chemical potential (as a function of $q$) is chosen to be that of a zero temperature spin-1 condensate in the easy-plane phase, which is obtained by solving $\mathcal{L}_m \psi_m = \mu \psi_m$. This gives [14],

$$\mu = g_n n_0 + g_s n_0 + \frac{q}{2}. \tag{18}$$

These choices of $T$ and $\mu$ give steady state energy and atom number within 1% of the conservative GPE results. We use $\gamma = 10^{-2}$, which gives $|dW| \lesssim q_0 |\psi_m| dt$, and therefore reservoir scattering events occur within the time scale of spin interactions. The microscopically derived value for $\gamma$ will be considerably smaller than this [53], resulting in the GPE dynamics overwhelming the reservoir interactions.

We evolve Eq. (14) using an interaction picture fourth order Runge-Kutta integrator with periodic boundary conditions and kinetic energy evaluated to spectral accuracy. The noise is added in a single step following the Runge-Kutta integration of the $(1 - i\gamma)(\mathcal{L}_m[\psi_m] - \mu \psi_m)$ term [60]. Numerical parameters and initial condition sampling are the same as for the GPE simulations.

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
