# Peer review of "Anomalous phase ordering of a quenched ferromagnetic superfluid"

_SciPost Physics, doi:SciPost Phys. 7, 029 (2019)_

## Round 3 · Referee Report · Anonymous (Referee 1) · 2019-7-3

Strengths

1) The authors found a new regime of coarsening, which needs long-time simulations. 2) They also found features of a novel turbulent cascade.

Report

The authors study the coarsening dynamics of a spin-1 condensate quenched into the easy-plane ferromagnetic phase. In the isolated case, after the coarsening driven by pair annihilation of spin vortices, slow coarsening associated with spin wave energy transport continues. They point out that the transport shows features of a spin wave energy cascade. They also study open system dynamics and show that the system thermalizes after the annihilation of vortices in that case.

I recommend to publish this manuscript in SciPost Physics. However, I have a few questions.

1) The total energy of spin waves decreases in the isolated system (Fig.2(c)), although the energy is conserved. Where did the rest of energy go?

2) It is sure that the $t^{1/3}$ behavior is a feature of the model B universality class. Isn't there any other universality classes that show $t^{1/3}$ behavior? Since the symmetry of the system is different from that of the model B, I am not sure if the model B should be mentioned in Conclusion.

Requested changes

1) The total energy of spin waves decreases, which is shown in Fig. 2(c). Mention what the decrement transformed into. 2) Discussion on the comparison with the model B should be revised.

---

## Round 4 · Author Response

We thank the referee for their constructive comments. We address the referee's points 1) and 2) below.

1) For clarity, we have renamed $E_\text{tot}$ to $E_\text{sw}$, and slightly reworded how this term is described, to make it clear that this quantity is only the spin wave contribution to the total energy, which is not a conserved quantity. The paper contains a description of where the lost energy goes, "There is also a net decrease in the total spin wave energy $E_\text{sw}$, showing that energy is also lost from the spin wave excitations, either to other quadratic excitations [36,46–48] or to excitations beyond quadratic order."

2) We have clarified in the Conclusion that ascribing our universal dynamics to the scalar model B dynamic universality class is only speculation. This is based on the scaling observed and the symmetry and dimension of the $nF_z$ field, which follows closely the dynamics of the order parameter. We have also left open the possibility for the dynamics to belong to a different dynamic universality class.

---

## Round 4 · List of Changes

1) For clarity, we have renamed $E_\text{tot}$ to $E_\text{sw}$, and slightly reworded how this term is described, to make it clear that this quantity is only the spin wave contribution to the total energy, which is not a conserved quantity. The paper contains a description of where the lost energy goes, "There is also a net decrease in the total spin wave energy $E_\text{sw}$, showing that energy is also lost from the spin wave excitations, either to other quadratic excitations [36,46–48] or to excitations beyond quadratic order."

2) We have clarified in the Conclusion that ascribing our universal dynamics to the scalar model B dynamic universality class is only speculation. This is based on the scaling observed and the symmetry and dimension of the $nF_z$ field, which follows closely the dynamics of the order parameter. We have also left open the possibility for the dynamics to belong to a different dynamic universality class.

---

## Editorial Decision

published